# Design of a Low-Cost Diffuse Optical Mammography System for Biomedical Image Processing in Breast Cancer Diagnosis

**DOI:** 10.3390/s23094390

**Published:** 2023-04-29

**Authors:** Josué D. Rivera-Fernández, Karen Roa-Tort, Suren Stolik, Alma Valor, Diego A. Fabila-Bustos, Gabriela de la Rosa, Macaria Hernández-Chávez, José M. de la Rosa-Vázquez

**Affiliations:** 1Laboratorio de Optomecatrónica, UPIIH, Instituto Politécnico Nacional, Distrito de Educación, Salud, Ciencia, Tecnología e Innovación, San Agustín Tlaxiaca 42162, Mexico; 2Laboratorio de Biofotónica, ESIME ZAC, Instituto Politécnico Nacional, Ciudad de Mexico 07320, Mexico; 3Hospital de Especialidades del niño y la Mujer Dr. Felipe Nuñez Lara, Santiago de Querétaro 76090, Mexico

**Keywords:** diffuse optical mammography (DOM), biophotonics, optical bioimaging, breast cancer diagnosis, image processing

## Abstract

Worldwide, breast cancer is the most common type of cancer that mainly affects women. Several diagnosis techniques based on optical instrumentation and image analysis have been developed, and these are commonly used in conjunction with conventional diagnostic devices such as mammographs, ultrasound, and magnetic resonance imaging of the breast. The cost of using these instruments is increasing, and developing countries, whose deaths indices due to breast cancer are high, cannot access conventional diagnostic methods and have even less access to newer techniques. Other studies, based on the analysis of images acquired by traditional methods, require high resolutions and knowledge of the origin of the captures in order to avoid errors. For this reason, the design of a low-cost diffuse optical mammography system for biomedical image processing in breast cancer diagnosis is presented. The system combines the acquisition of breast tissue photographs, diffuse optical reflectance (as a biophotonics technique), and the processing of digital images for the study and diagnosis of breast cancer. The system was developed in the form of a medical examination table with a 638 nm red-light source, using light-emitted diode technology (LED) and a low-cost web camera for the acquisition of breast tissue images. The system is automatic, and its control, through a graphical user interface (GUI), saves costs and allows for the subsequent analysis of images using a digital image-processing algorithm. The results obtained allow for the possibility of planning in vivo measurements. In addition, the acquisition of images every 30° around the breast tissue could be used in future research in order to perform a three-dimensional (3D) reconstruction and an analysis of the captures through deep learning techniques. These could be combined with virtual, augmented, or mixed reality environments to predict the position of tumors, increase the likelihood of a correct medical diagnosis, and develop a training system for specialists. Furthermore, the system allows for the possibility to develop analysis of optical characterization for new phantom studies in breast cancer diagnosis through bioimaging techniques.

## 1. Introduction

According to the World Health Organization (WHO), breast cancer is the most common type of cancer in women, and it affects all countries around the world [1]. It is the most frequent neoplasm, with an estimated 1,671,149 new cases per year, of which 521,907 end in death [2].

In Mexico, which is a developing country with poor access to reliable diagnostic systems, malignant breast cancer is the main cause of death in women, and only 15% of the detected cases are identified in the early stages [3]. In 2014, the mortality rate was 17.6 deaths per 100,000 women aged 25 years and over, and that number grew to 18 deaths per 100,000 in 2015 [4]. The northern region of Mexico and Mexico City had the highest breast cancer mortality [5,6].

Currently, the only accepted method for an early diagnosis, due to its effectiveness, is mammography. In developing countries such as Mexico, it is difficult to obtain these resources due to their high cost [1]. Moreover, the high cost of maintenance results in improperly calibrated equipment, leading to incorrect diagnoses [7]. Thus, clinical breast cancer studies are limited to the specialist’s experience or to auto-exploration [8], resulting in late diagnoses of the advanced stages of the disease. 

Due to the high cost and poor accessibility of existing high-tech diagnostic methods as well as due to the advantages and disadvantages of various modalities in breast imaging [9], new diagnostic methods are being studied, including the development of optical instrumentation systems such as diffuse optical mammography (DOM) [10].

DOM is a non-invasive monitoring technique for biological tissue which uses non-ionizing radiation in the therapeutic window [11]. DOM is a specific detection technique that aids medical diagnosis by identifying abnormalities in breast tissue. This technique has its origins in the first decades of the 20th century with the experiments of Max Cuttler [12], although, at that time, it was not known by the name DOM. In general, the idea is to perform the observation of the breast tissue internally using red- or near-infrared (NIR)- light sources to irradiate the tissue and obtain information on the optical properties and the diffuse reflection images of the tissue.

It is known that, in breast cancer, the tissue has a higher concentration of blood, which implies a greater absorption of red and NIR light. The tissue presents an increased amount of water (H_2_O) and a decreased lipid concentration compared to healthy breast tissue [13]. The tissue optical index (TOI) has been established as an indicator by which we can determine the malignancy of tissue based on hemoglobin (Hb), H_2_O, and lipid concentrations [14].

DOM is a technique that can be implemented in different ways, depending on the experimental arrangement. Four types can be identified: transillumination of the breast tissue, planar geometry, circular geometry, and reflection mode [10].

Transillumination of breast tissue could be considered the first kind of DOM, and it has been implemented since the early 1930s. This technique was known as vascularity shadows. The method involves illuminating breast tissue with a red-light source placed on the surface of the breast. Initially, transillumination was used specifically as a visual medical examination in a dark room, due to the low technological development that did not allow for obtaining quality images, and the light sources used a wide spectral width [12]. With the development of digital cameras and narrow-spectral-width lighting sources, the technique became easier to apply and presented better results in the early neoplastic stages [15,16,17]. 

The arrangement of the instrumentation for DOM by planar geometry is similar to traditional X-ray mammography, in that the breast is supported with two plates for compression holding. DOM by planar geometry differs from the traditional mammogram because the lighting source and the optical detectors are on one of the plates [18]. Analyzing the tissue through spectroscopy techniques in the frequency domain sometimes involves time-of-flight studies of light passing through tissue [19,20]. Moreover, implementations of DOM by the planar geometry technique have also been carried out on medical examination tables incorporating digital cameras, generally with a charge-coupled device (CCD sensor) for the acquisition of images [21].

DOM in a circular geometry configuration is performed by placing optical fibers around the breast when it is hanging, and it is generally implemented in a medical-examination-table-type system. This arrangement makes use of techniques in the frequency domain, and it is focused on obtaining information at different heights for analyzing the optical properties of the breast tissue for areas with a higher Hb concentration [22].

The reflection method is based on diffuse reflectance measurements using a handheld optical probe; in this case, the detection and illumination fibers are on the same breast side, and a single source–detector pair has been proposed [23]. This technique is very useful for locating tumors that are close to the surface of the breast, because the optical probe scans the surface and some millimeters inside the breast [24,25,26,27], but the principal problem is that the implementation of this instrumentation could be expensive.

### DOM System Design Requirements

According to Bigio et al. [10], there are four important design requirements to consider in the development of optical instrumentation for DOM systems. Firstly, the patient’s position during the analysis should be ergonomic, and this will determine the configuration of the DOM system with reference to the transillumination, planar, circular, or reflection modes. The second point is related to the optical, electronic, mechanical, and other physical elements, in other words, the hardware of the DOM system. After that, it is important to consider the method used to obtain the tissue information, for example, using a camera to take pictures, using an optical sensor to study the light propagation through the tissue, or both. The last point is to consider how to process the information (images or measurements of light) and how to display it as a medical diagnosis. In other words, the medical imaging technique or the signal processing that provides the diagnosis information must be considered [9,28]. Imaging performs a central role in the screening, diagnosis, and staging of breast cancer, as well as in post-treatment surveillance [29]. 

According to some research, methods based on optical techniques for the diagnosis of breast cancer have been explored as alternative options. However, the implementation of optical instrumentation, such as filters, optical fibers, laser-based light sources, and electronics, is expensive when it is used as a complement to conventional techniques; for developing countries, access to this type of instrumentation is not possible.

The present work describes the development of a low-cost DOM that uses non-ionizing radiation and permits specialists to give or complement their breast cancer medical diagnoses. Moreover, the discussed development could be an option for countries with poor access to current diagnostic techniques due to their cost. In addition, the system could be used not only for diagnosis, but also as a sensor to characterize phantoms for biomedical image-processing instrumentation.

## 2. Materials and Methods

The four points described above encompass the aspects to be considered for the development of optical instrumentation, but it is important to mention that these elements are strongly related which each other, so it is not easy to prioritize one with respect to the others. The elected design will depend on the required parameters, such as the precision of the system, the specificity or sensitivity in the diagnosis reports, and the resources available to develop the system.

### 2.1. DOM System

A complete diagram of the developed DOM system is depicted in Figure 1. As we can see, it is composed of four elements: the illumination source; the irradiated breast tissue (or a phantom); the hardware to capture the images or optical signals from tissue; and, finally, the computer tools which process the information obtained in order to provide or compliment a medical diagnosis.

The developed system is an automated medical examination table that uses the transillumination method applied around the entire breast to capture images from a digital camera; for that reason, the configuration will be called a three-dimensional (3D) transillumination system. Its functional modules are shown in Figure 2. The first one, identified as the hardware of the system, is composed of the mechanical, electronic, and optical elements required to operate the system. The second functional module, called the control and processing module, interacts with the hardware and the processed data. This module refers to the computational interfaces (software) which operate the system and acquire, store, and process the information collected from the tissue. Finally, the third functional module enables visualized information to provide or complement a medical diagnosis.

The hardware is divided into two subsystems. The first is the mechanical subsystem, which includes the transmission devices, the supports for each element, and the medical examination table itself. The second is the electronic subsystem that controls the motors, the light sources, and the interface with the computer from which the system is operated. 

A medical-examination-table-configuration system is used to make the study easier and more comfortable. The proposed medical examination table has the standard dimensions of a traditional examination table with the distinction of having a cavity at breast height; when the patient is in a prone position, the breasts remain hanging inside the cavity. 

The key part of the system is the diagnostic plate, where the light source and digital camera are placed. The DOM system has two diagnosis plates so that it can analyze the tissues of both breasts separately.

### 2.2. Diagnosis Plates

The diagnosis plate, shown in Figure 3, is made of polymethyl methacrylate (PMMA) and is composed of two pieces joined by spacers. The lower part contains a normalized stepper motor by the National Electrical Manufacturers Association (NEMA), with 3.2 kg/cm holding torque and 1.8° per step (SY42STH47-1206A, POLOLU©), to rotate the plate around the breast. It supports a gear rack and pinion mechanism that allows the light source to move closer or further away. A digital camera is located on the opposite side of the plate and in front of the light source. The camera and the light source each have a support that maintains their vertical positions and avoids any disturbance during the rotation of the plate and the linear movement of the light source. Each support consists of a post to raise the optical sensors and place them to detect the presence of tissue. The upper piece works as a cover so that the mechanisms and supports are not in contact with the breast tissue. 

The device meant to move the plate is an electronic linear actuator (LAD-B8-BK, MCP™) placed at the center of the plate. To raise it or to return it to its initial position, and to give better stability to the plate, two linear guides with displacement carriages are used as support elements.

To design the gear rack and pinion mechanism, the following aspects were considered: simple structure, high rigidity, small size, lightweight, and excellent responsiveness [30]. Both pieces must have the same pitch, P; the same tooth modulus, m; and the same pressure angle, α. This last value is a standard value that could be 20° or 25°. In this case, the values (P, m, α) are (20, 1, 20°), respectively.

It is important to mention that all the mechanical systems implemented in the developed instrumentation have sensors to locate the dimension limits of the system in order to provide a safety system for patients. 

### 2.3. Light Source

Because Hb shows a high absorption in the red spectral region [31], a red-light emission diode (LED) (Siled^®^ LED-P1R55-120/41) was selected as the light source of the transillumination system in order to reduce system costs. The emission peak of this LED is a wavelength of 638 nm, and it has a full width at half maximum (FWHM) of 21 nm. Its wavelength shift is less than 1 nm when the temperature increases from 25 °C to 27 °C. The LED was mounted on a heatsink to hold the required operating temperature at 25 °C and to avoid burning the patient. It is important to highlight that the LED light source was used instead of laser to avoid damage to the patient during the irradiation process. The temperature was maintained at 25 °C to avoid photothermal effects.

The irradiation power of the light source is variable and is modulated by a pulse-width modulation (PWM) signal, whose operating frequency is high enough to capture the images. Using a constant-current LED driver with PWM dimming, it is possible to adjust the different values of the irradiation power in a linear function (see Figure 4).

The measured power has a minimum value of 4.4 ± 0.2 mW, at 10% of the duty cycle, and a maximum of 40.9 ± 0.2 mW, at 100% of the duty cycle.

### 2.4. Digital Camera

For image acquisition in the DOM system, a web camera (Logitech© C170) was selected due to its low cost and small dimensions. Table 1 shows its principal characteristics.

The camera was characterized to determine the distance at which the image would have good contrast and to obtain the standard deviation per pixel after capturing several images. For this case, one hundred images were captured, and a pattern, or base image, was obtained, which eliminated the intrinsic noise of the device. The characterization, where a pattern image is calculated to avoid the device noise, is sometimes performed using the mean squared error (MSE) and the peak signal-to-noise ratio (PSNR). The MSE represents the cumulative squared error between the images, whereas the PSNR represents a measure of the peak. The PSNR is used as a quality parameter between two images. The higher the PSNR, the better the quality. The lower the MSE, the lower the error [33].

### 2.5. Control Interface

To operate the developed system, a connection between the hardware and the software, a computer control, is necessary. For this, the Microchip^®^ ATmega2560 AVR microcontroller was used as the central unit. It activates the electronic control circuits necessary for the operation of the system using the signals received from a graphical user interface (GUI) installed in a computer.

The GUI was developed using the *Appdesigner* module from MATLAB^®^ software. Figure 5 shows the main interface screen. It consists of three main sections—control, visualization, and breast diagram—as well as an area for the patient’s name as an indicator of the captured images.

The control area involves five buttons, a slider bar, and an indicator. This area is located on the left side of the main interface window and labeled as right breast or left breast controls. This area is responsible for manipulating the optical, electronic, and mechanical elements of the system. 

The control buttons are used to establish the parameters and to operate the linear actuators that move the diagnostic plates vertically (Up and Down buttons), the gear rack mechanism responsible for zooming the lighting in/out (Zoom In and Zoom Out buttons), the motor that controls the rotation around the breast, and the button that starts the automatic action to capture images from the breast tissue (Start button). The slider bar controls the pulse-width modulation of the LED driver to establish, based on the PWM signal, the irradiation power of the light source.

The camera image is located at the center of the interface; this setup was chosen because it is easy to observe the progression of the study in the foreground [34]. The displayed image is always the last one; the displayed image is updated each 30° until the capture range of the breast is completed.

The images are captured according to the breast quadrants [35]. This is a medical technique that divides the breast using the nipple as a reference point, forming four divisions or quadrants—(the upper internal quadrant (UIQ), the upper external quadrant (UEQ), the lower external quadrant (LEQ), and the lower internal quadrant (LIQ))—to determine where the lesion is located. Each image shot was performed every 30° around the tissue.

When the software is initialized in the breast diagram area, it shows both breasts divided into quadrants, as in Figure 6a. The conforming images are saved, and the diagram changes to indicate the capture position, as shown in Figure 6b. 

At 30°, the diagram corresponds to the 1 o’clock breast position at 60°, the image would represent 2 o’clock, and so on until the breast study is completed.

It is important to mention that the division into quadrants is a technique used by specialists to determine or approximate the location of tumors. In the case of the developed system, the location of the nipple is not decisive, since the patient is lying down in a prone position and the camera is oriented in a fixed initial position aligned to the posture of the patient (12 o’clock). Although this could generate a deviation in identifying the tumor location, due to poor posture of the patient on the medical examination table, this variation is not significant considering that a future study on 3D reconstruction of the tissue will be developed.

Finally, in the lower part of the window, there is an entry box to introduce a patient’s name or an ID, and, next, a button to exit the application.

### 2.6. Phantom Tissue

Polydimethylsiloxane (PDMS), commercially known as “silicone breast implants,” are the most common phantoms currently in use [36]. They are similar, in dimensions and form, to real breasts, and can be used as phantoms in the study of abnormalities in breast tissue [37,38]. We used commercially available implants from POLYTECH Health & Aesthetics GmbH Company. The phantom was introduced with India ink to create spheres (with a diameter between 6 mm and 10 mm) to simulate cancer, and with PVC plastic wires (0.5 mm and 1.2 mm diameter) to simulate veins and blood vessels. 

For the first stage, the goal of the phantom is to determine the minimum size that the system can detect for the digital processing of biomedical images. The decision was made to implement the polydimethylsiloxane silicone-based phantom because it has similar mechanical and optical properties to real breast tissue. These kinds of phantoms are used to test systems that implement diffuse optical imaging with red or NIR light for biomedical image processing in breast cancer diagnosis [39]. 

India ink’s optical properties are known to be similar to a tumor in the red and NIR irradiation range [40,41]. Considering that the developed system uses a wavelength of 638 nm, India ink is an excellent material with which to simulate tumors. 

On the other hand, the wires are only used to provide a pattern of known dimensions to serve as a reference for the minimum size that the developed system can detect. In the first stage, the phantoms were handled separately, that is, one for the simulation of tumors and the other one for the simulation of veins and blood vessels. Subsequently, we intend to combine these elements and even use other types of phantoms. This would allow for the possibility of using the system for the characterization of the phantom properties used in the diagnosis of breast cancer through biomedical image processing.

## 3. Results

The aim of this investigation is to make a low-cost DOM system. The reported results show the characterization of the system and the initial tests for the breast phantom studies.

### 3.1. Mechanical DOM System

The configuration of the developed DOM system is based on a traditional medical examination table, but with the variant of a cavity for the study of breast tissue, which is shown in Figure 7. The cavity dimensions, when viewing the table from left to right, are 250 mm in length and 500 mm in width, and they were chosen considering the anthropomorphic sizes of Mexican women [42]. Inside the medical examination table are (Figure 7a) the electronic components (control, power, and supply circuits; Figure 7b) and the mechanical and optical components of the two diagnostic plates (Figure 7c).

### 3.2. Electrical Characterization

For the system characterization, we identified the stability of the operating electrical current per 10% of the duty cycle. Table 2 shows the values for each measurement, and Figure 8 shows the linear fit that represents the increase in the LED operating current as a function of the PWM signal. 

The electrical power as a function of the PWM signal, shown in Figure 8, adjusts to a straight line, and it should be noted that the uncertainties are in the order of µA, which indicates that the LED driver is stable during system operation. 

The electrical current as a time function is shown in Figure 9 and indicates that the uncertainty in most of the measurements was less than 30 µA, except for a duty cycle of 70%, where the uncertainty measure was 70 µA. Nevertheless, these values are not significant for the operation of the system. The relative uncertainties in the current are less than 0.1 mA.

Finally, the irradiation power as a function of the operating current is shown in Figure 10.

### 3.3. Test on Phantom Tissue

According to the American Cancer Society, breast cancer is classified with a “T” followed by a number, which ranges from 0 to 4, to describe the main (primary) tumor’s size and if it has spread to the skin or to the chest wall under the breast. Higher T numbers mean a larger tumor and/or wider spread to tissues near the breast. When a tumor is 20 mm or less in diameter, it is a T1 level tumor [43]. In addition, internal breast vessel diameters are around 2.5 mm [44]. 

For the first tests in the developed DOM system, two breast prostheses from POLYTECH Health & Aesthetics GmbH Co. were used as phantom tissue in two different configurations, as shown in Figure 11. The first configuration, Figure 11a, simulates spherical abnormalities (6 and 10 mm diameter) inside the tissue using India ink, while the second one, Figure 11b, was used to simulate veins and blood vessels (0.5 and 1.2 mm diameter) in the tissue using polyvinyl chloride (PVC) plastic wires.

The diameter of the simulated tumors and the blood vessels is crucial because patients have a higher probability of survival when the findings inside the tissue are less than 20 mm, which implies that patients are in the early stages of the pathology [45]. 

To simulate smaller tumors, it has to be considered that, by changing the study from the phantom to an in vivo study with patients, the detection of abnormalities or vascularity will be of lower resolution [46]. Through the dimensions of the simulated tumors and the vascularity, a known pattern could be generated to estimate the actual size of a tumor in vivo at the time of processing the captured images.

The image processing algorithm consists of three distinct stages. First, the green channel of the image is selected, which helps to contrast the vascularity inside the tissue due to the optical properties of hemoglobin. This technique is known as “red-free” [47,48,49]. The second stage consists of obtaining the inverse of the green channel to segment the contours. Finally, the third stage contrasts the contours in the image with the original capture. Figure 12 shows a block diagram of the algorithm used to process the digital images.

A test with the prosthesis that simulates tumors was carried out by processing the image in MATLAB^®^ to identify the area of abnormality and to subsequently perform a segmentation of the area of interest, as shown in Figure 13. After the tests on the tumor phantom, the same process was carried out for the veins and blood vessel phantom. This information is shown in Figure 14.

In the phantom tumor images, the contrast between the contour image with the original image, as in Figure 13d, is not clearly observed; i.e., the abnormalities detected by the system are not completely segmented. This occurs because the camera detects the reflection from the light source. This phenomenon does not represent a drawback, because it is totally dependent on the phantom’s material, whereas, in in vivo studies, the images will only depend on breast density and the patient’s skin color.

In Figure 13d and Figure 14d, the transmitted light of the irradiation source can be appreciated, but, in Figure 13d, the segmentation of the findings inside the phantom shows better results. This is because the reflection is not close to the interest area.

It is possible to distinguish the different elements inside the phantom (tumors or vascularity), but, due to lighting, it detects the transmitted light of the source as noise because of the contrast between the background and the objects of interest; therefore, when the algorithm identifies the contours, the transmitted light is also detected. Considering that the goal of the developed DOM system is to provide visual reference information that allows specialists to give or complement a diagnosis, we decided to show, as additional information, the image that represents the inverse of the green channel, where the tumors or vascularity inside the phantom are displayed in a similar way as in conventional mammography.

Being able to distinguish the images of both the simulation of tumors and the simulation of vascularity inside the phantoms, as a first approximation, is a favorable result, but carrying out future tests with different kinds of phantoms to improve the system is not ruled out. It is important to highlight that the simulated dimensions are in an interval consistent with the measurements indicated in the initial stage of the disease (T1) [43,44].

If several kinds of phantoms are implemented, it is possible that these can be characterized in situ in the system. This could indicate whether a greater number of lighting sources is required, whether a different wavelength is necessary to achieve greater penetration in the system, or if changes in the density and other mechanical properties of the phantom could lead to a better simulation of the tissue properties.

### 3.4. Validation of the Image Acquisition System

To quantify the system’s operation, the images were numerically analyzed using the MSE and the PSNR. For this, one hundred captures were taken and compared in each stage of the algorithm (green channel, green channel inverted, and image of segmented contours) for the same captured area. MSE and the PSNR were calculated through an average value for the captures, considering the first shot as a reference, thus determining which stage produced the better image and gave the information required in order to obtain the best diagnosis. The results are shown in Table 3.

As Table 3 shows, it is possible that the stage of the algorithm that produces the inverse of the green channel provides a better image, which confirms the fact that the reflection on the phantom does not have the same effect when using the inverse of the green channel compared to the other stages of the algorithm. Despite the differences between each stage which were expected, this variation is minor. In general, the selected algorithm is adequate to obtain images of the abnormalities inside the phantom. To prove that the radiation source reflection will not act in the same way on a patient’s tissue as it does in the phantoms, it will be necessary to implement the system in a protocol for in vivo studies.

## 4. Discussion

Currently, there are various medical diagnostic devices for breast cancer detection. Nevertheless, in developing countries, the high cost of diagnosis and access to such devices can prove to be a challenge. On top of that, the maintenance of these devices does not follow the regulatory standards and is inadequate for providing a correct diagnosis. In such circumstances, the diagnosis depends on the specialist’s experience or on techniques such as self-examination, making the diagnosis possible only in the advanced stages of the disease [1,7].

The hypothesis underlying the development of this low-cost system is that the use of digital cameras, image processing, and LED lighting sources placed on the surface of the breast or close to it, implemented to study the propagation of light using image-processing and optical techniques, allows the characterization of breast tissue to detect areas of vascularization or possible abnormalities that aid the specialist in the diagnosis of breast cancer.

In the proposed DOM system, we decided to implement two diagnostic plates for the study of both breasts. The DOM system has an LED as an illumination source in each diagnostic plate, whose emission peak is centered at 638 nm, with an FWHM of 21 nm. Each LED is powered by a current source controlled by a PWM signal, thus obtaining 10 intervals of radiant intensity, i.e., an interval every 10% of the useful duty cycle of the PWM signal, where 4.40 ± 0.5 mW is the minimum value and 40.90 ± 0.2 mW is the maximum value. The interval was selected to control the irradiation power considering the patient’s skin characteristics. The LED power was supplied at 12 V with a maximum current operation of 226.57 ± 0.03 mA. The system has a loss of 1 mW of irradiation power due to the variation of 30 µA in electrical consumption; this loss occurs when the device is working continuously for 10 min, but it is insignificant since it does not influence the obtained results.

The GUI consists of three main areas. The first one is the control area, where the parameter configuration of the electronic and mechanical components of the system is located. The second one is focused on the visualization of the captures that are made in the study. The last one is for the visualization of the area of the last capture. The acquisition of digital images is automatically performed every five seconds. These images are processed using an algorithm developed in MATLAB^®^, which can identify abnormalities in the phantom with a minimum diameter of 0.5 mm. It is necessary to create a protocol for the implementation of the system in patients to determine the real size of abnormalities that can be detected by the system; however, the results obtained in the phantoms are promising for future works.

From the images shown in Figure 13 and Figure 14, there are favorable results in the sense of being able to identify areas of interest in breast tissue using diffuse optical mammography techniques and the implementation of low-cost optical, electronic, and mechanical components in the automation of a system for the study or analysis of the breast. The images captured by the DOM system allow for the identification of abnormalities in a phantom and may be viable for the development of an algorithm that combines the images and generates a 3D arrangement from the two-dimensional information captured in each image. This will provide additional information for identifying the position of the tumor or abnormality, as well as the tissue vascularity. Implementing a 3D model would make it possible to generate a guide to locate the position of the abnormality for surgical intervention and as auxiliary visual information for specialists. The latter is an advantage of the system when compared to other low-cost systems. At the time of delivering a diagnostic report, it is possible to complement these results using quantitative methods from the optical properties of the breast tissue and the elements used here.

Regarding the design of the physical part of the system, the mechanical system provides a stable and comfortable position for patients when they are placed in a prone position. The use of two diagnostic plates, instead of one, makes the individual study of each breast possible. It should be noted that, in the design and construction, the dimensions of the DOM system were chosen considering Mexican women’s anthropomorphic sizes [42]. 

A 5-megapixel fixed-focus digital camera was selected to capture images for the purpose of identifying abnormalities or vascularity within the phantoms. It is possible to improve the image resolution by using a better digital camera, such as an infrared camera, combined with deep learning techniques to obtain more useful information for the diagnosis [50].

### 4.1. Comparison with Other Diagnostic Systems through Optical Techniques and Digital Image Processing Based on Low-Cost Systems

Several works where conventional mammographs are combined with photonic instrumentation for their operation could be mentioned. Packyanathan et al. developed a diffuse optical tomography system using NIR wavelengths [48]. Unlike the proposed system, this one required pressing the tissue, as a conventional mammograph does, while the proposed system seeks to not press the tissue and to avoid discomfort for the patient by only irradiating the tissue and not touching or pressing it.

Chae et al. reported a system where the reconstruction of the shape of the breast tissue was performed [20]. This system requires the use of more than one camera to generate a 3D image of the tissue. In contrast, the system presented in this manuscript makes use of a single low-cost camera that rotates around the breast tissue and is able to take photos every 30° without having direct contact with it. These could be used in future reconstruction analyses using point-of-similarity techniques between the images, as performed by 3D scanners.

Systems based on thermography or infrared images [51,52] analyze the temperature change in the breast tissue or its segmentation, respectively, to determine abnormalities through digital image processing. Despite this, tumor location is not directly obtained with these techniques, and they require the application of other techniques. In addition, these methods usually do not have a high accuracy in the early stages of the disease. Imaging techniques normally show qualitative information, so they need other methods to be able to optically characterize the tissue. With the proposed system, the properties of different phantoms (mainly optical properties) could be characterized and in vivo diagnoses could be carried out.

Several authors have reported the use of machine learning or deep learning techniques, such as neural networks or classification systems, to analyze images obtained from studies of conventional diagnostic methods [20,50,51,52]. The results show a high efficiency and clear classification in differentiating between healthy and abnormal tissue when the databases have a great number of images and high resolution. Unfortunately, the origin of the captures or the characteristics of the image acquisition system are not always exactly known; consequently, it is important to determine the images and the specifications of the acquisition system before carrying out the training algorithms. The proposed DOM system realizes the image capture and storage to generate useful databases for subsequent investigations, with the advantage that the characteristics of the acquisition system are known, which provides data control for a continuous patient analysis over a period of time.

Several commercial instrumentation systems are similar to this proposed DOM system. They include red-light manual illumination sources (630–638 nm) [53,54], which are used to irradiate the breast tissue in a dark room and auscultate the patients. Notably, they do not have an image capture or acquisition system; for this reason, this kind of system cannot make an evolutionary comparison for the diagnosis or treatment of a patient unless a second person takes pictures with some external camera. In the proposed system, the automatic acquisition and digital processing of images allows the possibility of obtaining an evolutionary diagnosis or a complement to the examination carried out by the specialist.

### 4.2. A Brief Look at Potential Follow-Up Research Studies

With the development of the proposed system, the next step in the analysis of breast tissue is to use the acquired images to generate a reconstruction of the breast using point-matching techniques and machine learning or deep learning techniques. This could be used to develop 3D models to visualize the tissue and allow its analysis through virtual, augmented, and mixed reality environments in order to locate the abnormalities within the breast tissue or to generate training systems for medical students.

Regarding phantom characterization, it is necessary to implement different types of phantoms, since it is known that the tissue is a complex, non-homogeneous medium, and it must be considered that every patient has different skin colors and different densities of breast tissue. On that account, the latter is important in the second stage of tests for several types of phantoms.

Tumors and blood vessels were simulated separately in this first stage because the principal aim of this research was to determine the minimum abnormality dimensions that could be detected inside the phantom by the system, so we intend to use a phantom with both elements to verify that the system is capable of differentiating between them.

With the above, other types of studies and the implementation of various metrics for the use of the system in the characterization of breast tissue phantoms and in vivo studies could be extended into research on implementing 3D image acquisition systems, such as time-of-flight cameras or similar techniques, considering that these types of devices are now more affordable. Additionally, for follow-up research studies, by incorporating a greater number of measurements, where metrics such as macro and micro averages are added, as well as considering more parameters in order to achieve a mannequin with a greater similarity to biological tissue, the system can be improved while maintaining its low cost.

## 5. Conclusions

The design of a low-cost diffuse optical mammography system for biomedical image processing in breast cancer diagnosis was presented, where the aim of the work was to propose an automated system for the study of the breast tissue using the transillumination technique. It has two diagnostic plates, one per breast, with the intention to make studies easier and more comfortable.

The system was used to observe India ink which simulated spherical abnormalities between 6 and 10 mm in diameter, as well as to observe the simulation of veins and blood vessels from 0.5 to 1.2 mm in diameter, inside the phantom. Considering that a T1-stage diagnosis occurs when the tumor has a maximum diameter of 20 mm, and considering that the average size of the blood vessels in breast tissue is 2.5 mm, the system is viable for implementation. The next stage consists of simulating a phantom that combines tumors and blood vessels, as well as other types of phantoms that consider a greater number of the characteristics or properties of biological tissue, including the patient’s skin color.

The results provide the possibility for potential follow-up research studies to implement in vivo measurements to assess the instrument’s performance in operational conditions. In addition, this system could be used to generate a 3D model of breast tissue using virtual, augmented, or mixed reality combined with deep or machine learning algorithms. This could be used to predict the position of tumors for diagnosis or surgery. In addition, the system is useful for developing and characterizing phantoms for biomedical image processing in breast cancer diagnosis.

## Figures and Tables

**Figure 1 sensors-23-04390-f001:**
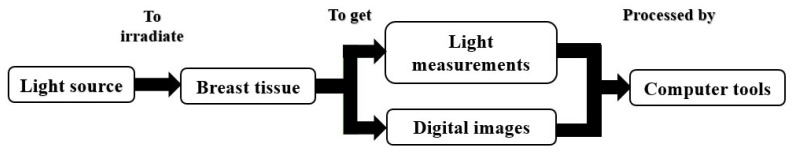
Block diagram of the diffuse optical mammography elements.

**Figure 2 sensors-23-04390-f002:**
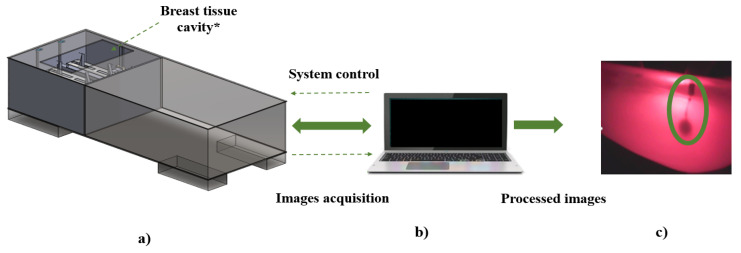
Functional areas of developed DOM system: (**a**) hardware, (**b**) control, and (**c**) processing. * Dimensions are according to Mexican women’s breast tissue standard sizes.

**Figure 3 sensors-23-04390-f003:**
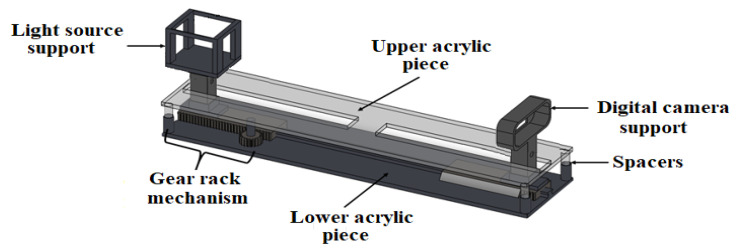
Diagnosis plate design.

**Figure 4 sensors-23-04390-f004:**
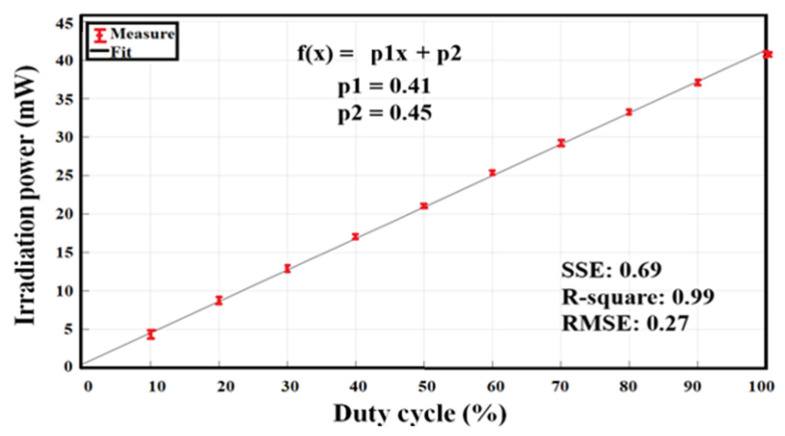
Linear fit of irradiation power of the light source as a function of the PWM duty cycle signal.

**Figure 5 sensors-23-04390-f005:**
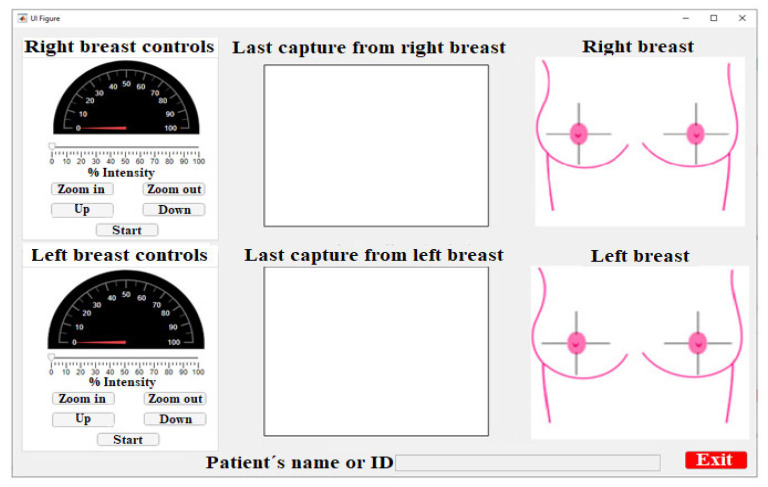
Control interface main screen for the developed DOM system.

**Figure 6 sensors-23-04390-f006:**
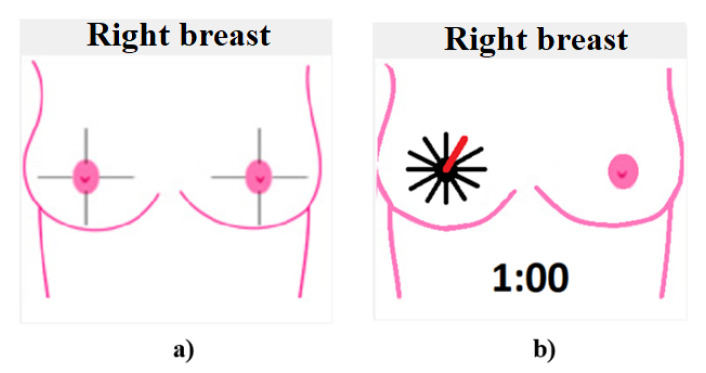
Breast diagrams: (**a**) quadrants of the breast, and (**b**) capture of the first position shot at 30°.

**Figure 7 sensors-23-04390-f007:**
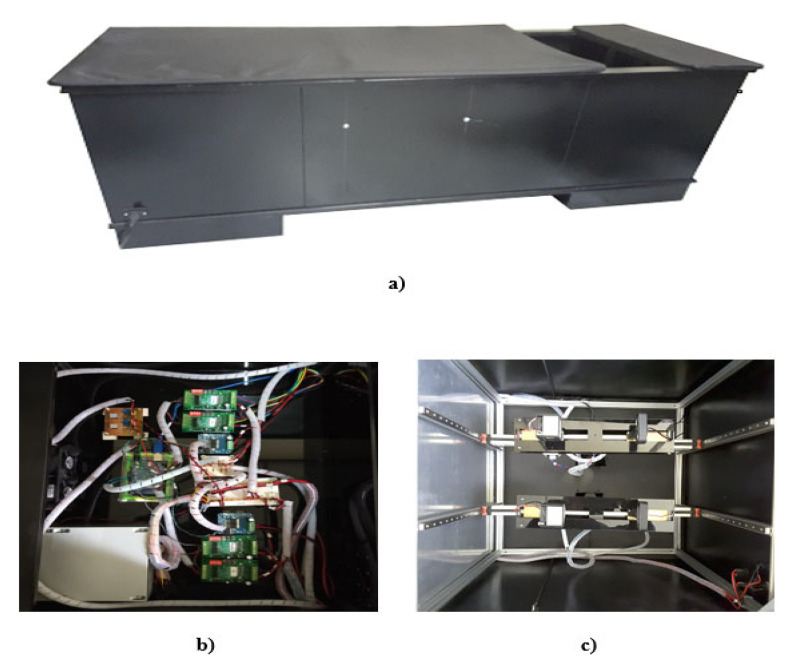
DOM system: (**a**) medical examination table configuration with a breast cavity; (**b**) electronic elements: control, supply, and power circuits; and (**c**) diagnosis plates with its optical and mechanical components.

**Figure 8 sensors-23-04390-f008:**
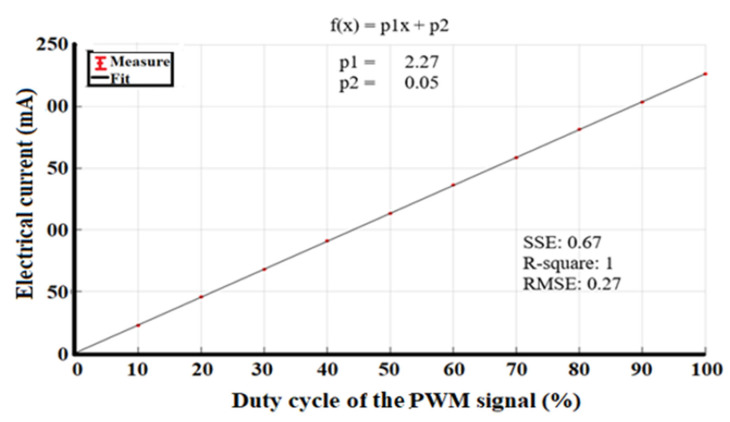
Linear fit of electrical current as a function of the PWM duty cycle signal.

**Figure 9 sensors-23-04390-f009:**
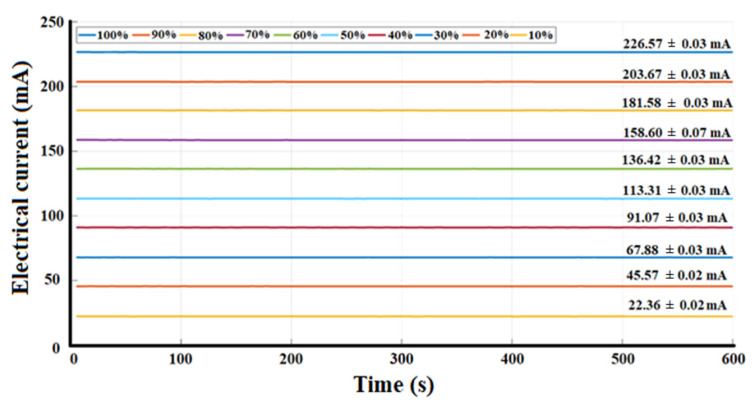
Electrical current stability as a time function.

**Figure 10 sensors-23-04390-f010:**
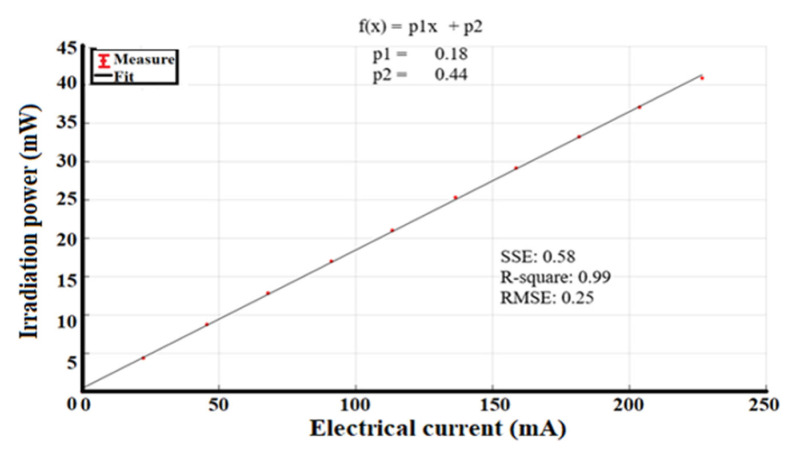
Variation of irradiation power as a function of the electrical current.

**Figure 11 sensors-23-04390-f011:**
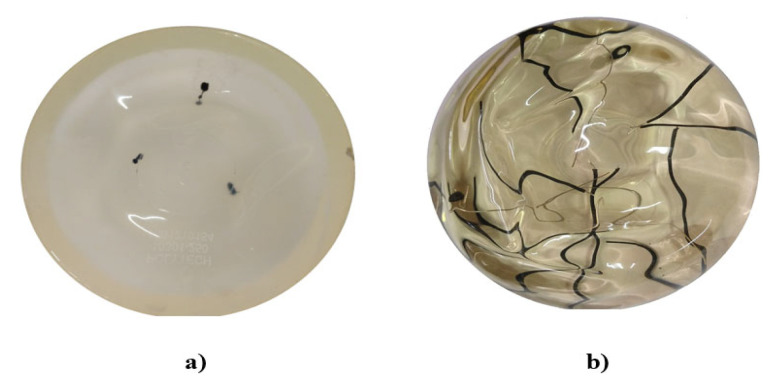
Phantoms: (**a**) tumor phantom using Indian ink, (**b**) vein and blood vessel phantom.

**Figure 12 sensors-23-04390-f012:**
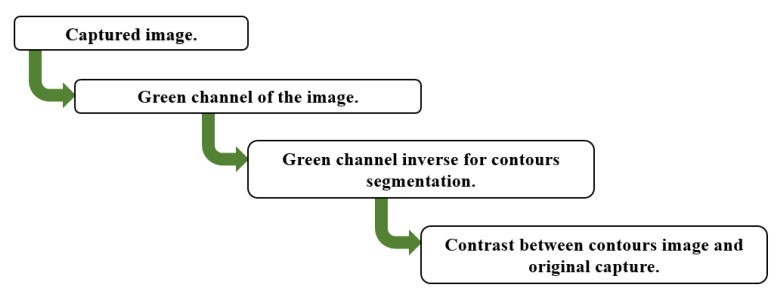
Image processing algorithm.

**Figure 13 sensors-23-04390-f013:**
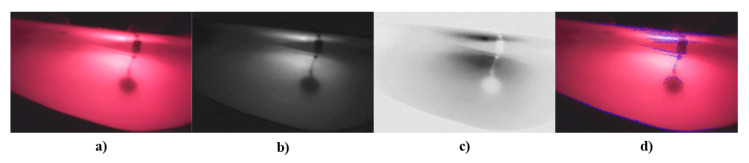
Results from tumor phantom: (**a**) original image, (**b**) green channel of the image, (**c**) inverse of green channel, and (**d**) contrast between contour image detection and original image.

**Figure 14 sensors-23-04390-f014:**
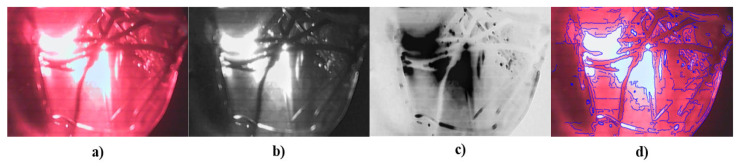
Results from blood vessel phantom: (**a**) original image, (**b**) green channel of the image, (**c**) inverse of green channel, and (**d**) contrast between contour image detection and original image.

**Table 1 sensors-23-04390-t001:** Camera principal specification [32].

Parameter	Description
Resolution	5 megapixels
Lens type	Standard
Focus type	Auto-adjustable
Visual field	60°
Communication protocol	USB

**Table 2 sensors-23-04390-t002:** Electrical current measurement as a function of the duty cycle of the PWM signal.

Duty Cycle	Electrical Current [mA]	Duty Cycle	Electrical Current [mA]
10%	22.36 ± 0.02	60%	136.42 ± 0.03
20%	45.57 ± 0.02	70%	158.60 ± 0.07
30%	67.88 ± 0.03	80%	181.58 ± 0.03
40%	91.07 ± 0.03	90%	203.67 ± 0.03
50%	113.31 ± 0.03	100%	226.57 ± 0.03

**Table 3 sensors-23-04390-t003:** Values of MSE and PSNR of 100 captures using the DOM system.

Algorithm Stage	MSE	PSN
Green channel image	4.81 ± 0.23	41.32 ± 0.25
Green channel inverse image	4.66 ± 0.21	41.45 ± 0.20
Contour segmentation image	4.74 ± 0.23	41.38 ± 0.21

## Data Availability

Not applicable.

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
