# Peer review of "Design of a Low-Cost Diffuse Optical Mammography System for Biomedical Image Processing in Breast Cancer Diagnosis"

_sensors, 2023, doi:10.3390/s23094390_

Round 1

Reviewer 1 Report

This manuscript had presented a low-cost diffuse optical mammography system for diagnosing breast cancer.  This paper is interesting for the oncology community. But some major comments that author should mentioned them:

1. Reorganize the abstract to conclude:
(a) The overall purpose of the study and the research problems you investigated.
(b) Major findings or trends found as a result of the study.
(c) A brief summary of your interpretations and conclusions.

2. Very few related works have been considered. Authors are advised to review and discuss the recent works related to breast cancer diagnosis (separate section).

3. The introduction needs to clarify the (1) motivation, (2) challenges, (3) contribution, (4) objectives, and (5) significance/implication. All the information should be presented sequentially.

4. More measures  (E.g. Also add metrics such as macro and micro average) and more parameters needs to be considered apart from the existing used ones.

5. The comparative analysis should testify how this work is more advanced?

6. The language of whole manuscript should be improved.

7.  Provide a brief look at potential follow-up research studies.

8. Conclusion should include the most significant findings and their implications.

9. Define all the symbols and notations used in the work. 

Reviewer 2 Report

The authors have presented an automated diffuse optical mammography system that utilizes low-cost components such as digital cameras, light-emitting diodes, and image processing techniques to generate images of breast tissue. The system is designed as an auscultation table with a pocket for breast tissue, and it uses a transillumination technique to capture images. The acquired images are processed using a "red-free" technique to detect abnormalities that could help in the diagnosis of breast cancer. The method and discussion need further improvement to be considered for publication.

1.       The safety considerations were not clear, although it was mentioned that a heatsink was used for the laser. However, this does not rule out the possibility that photothermal effect in the phantom or future patient samples. The irradiation power (~30 mW) is pretty high for skin contact for which the safe laser power would be lower than 1 mW.

2.       The author mentioned that the measurement is done by dividing the breast into quadrants with respect to the nipple. How was the location of the nipple calibrated?

3.       The phantom used ink and wire to simulate tumor and blood vessels, respectively. What are the justification of doing so? Do they have the same optical constants over the 633 nm regime?

4.       The ink and wire are added to two separate phantoms. Have the authors tried phantom tissues with both ink and wire? Do we expect the proposed DOM to be capable of distinguishing tumor and vessel? As the author mentioned in the text that ‘In in-vivo studies the images only will depend on breast density and the patient's skin color.’ Does the author consider to measure phantoms with different coloration and density to better mimic patient?

5.       The scientific presentation of this manuscript needs to be further improved to meet the standard of Sensor journal. A few abbreviations are not specified throughout the manuscript, such as Hb and PWM. There are also places with grammatic mistakes that requires the authors’ attention. For instance, the author writes ‘It is used an auscultation table configuration system with the intention to provide a 145 comfortable and easier study.’ Figure caption on page 9 is incorrect.

Overall, more details on the testing process and the results obtained should be provided to establish the validity of the system for detecting breast cancer. Furthermore, the paper could benefit from a comparison of the proposed system with other available low-cost diagnostic methods.

Reviewer 3 Report

The authors proposed an automated diffuse optical mammography system in the paper. It seems novel but I have the following questions:

1. The introduction refers to the "early stage", is the system suitable for early-stage diagnosis? There is no related illustration.

2. About diffuse optical mammography (DOM), its main characteristic is cheap, but the authors also mentioned low cost. But there is no comparison between the authors' and the existing devices.

3. The authors adopt LED lays at 638 wavelengths as a light source. The phantom is transparent, but as we know the human body has a complex tissue structure, such as skin and fat, which will affect the penetration of light. How solve this problem of penetration be applied to the human body?

4. The experiments did not support the low-cost statement.

5. There are grammar errors in the paper.

Round 2

Reviewer 2 Report

The authors have responded to my comments. Before the manuscript to be submitted, I would encourage the authors to do another round of final checks of the details. The labels of the figure are still not corrected, e.g., there are two figure 1 and two figure 2.

Reviewer 3 Report

The authors responded to my questions in detail. I have no other questions.
